# 4-*O*-Methylhonokiol Influences Normal Cardiovascular Development in Medaka Embryo

**DOI:** 10.3390/molecules24030475

**Published:** 2019-01-29

**Authors:** Santu K. Singha, Ilias Muhammad, Mohamed Ali Ibrahim, Mei Wang, Nicole M. Ashpole, Zia Shariat-Madar

**Affiliations:** 1Department of Biomolecular Sciences, Division of Pharmacology, University of Mississippi, University, MS 38677, USA; sksingha@go.olemiss.edu (S.K.S.); nmashpol@olemiss.edu (N.M.A.); 2The National Center for Natural Products Research, Research Institute of Pharmaceutical Sciences, University of Mississippi, University, MS 38677, USA; milias@olemiss.edu (I.M.); mmibrahi@olemiss.edu (M.A.I.); meiwang@olemiss.edu (M.W.); 3Chemistry of Natural Compounds Department, National Research Centre, Dokki-Giza 12622, Egypt; 4Light Microscopy Core, University of Mississippi, University, MS 38677, USA

**Keywords:** cardiomyogenesis, factor VII, factor X, inflammation, thrombosis, vasculogenesis, herbal medicine

## Abstract

Although 4-*O*-Methylhonokiol (MH) effects on neuronal and immune cells have been established, it is still unclear whether MH can cause a change in the structure and function of the cardiovascular system. The overarching goal of this study was to evaluate the effects of MH, isolated from *Magnolia grandiflora*, on the development of the heart and vasculature in a Japanese medaka model in vivo to predict human health risks. We analyzed the toxicity of MH in different life-stages of medaka embryos. MH uptake into medaka embryos was quantified. The LC_50_ of two different exposure windows (stages 9–36 (0–6 days post fertilization (dpf)) and 25–36 (2–6 dpf)) were 5.3 ± 0.1 μM and 9.9 ± 0.2 μM. Survival, deformities, days to hatch, and larval locomotor response were quantified. Wnt 1 was overexpressed in MH-treated embryos indicating deregulation of the Wnt signaling pathway, which was associated with spinal and cardiac ventricle deformities. Overexpression of major proinflammatory mediators and biomarkers of the heart were detected. Our results indicated that the differential sensitivity of MH in the embryos was developmental stage-specific. Furthermore, this study demonstrated that certain molecules can serve as promising markers at the transcriptional and phenotypical levels, responding to absorption of MH in the developing embryo.

## 1. Introduction

Magnolia bark extract has been used as a component of dietary supplements and cosmetic products [1]. One specific compound found in *Magnolia* species, 4-*O*-Methylhonokiol (MH), is recognized to have multifunctional activities both in vitro and in vivo, similarly to other honokiol analogs [2,3]. Magnolias appear to naturally produce a significant amount of these biphenyl-type neolignan compounds, many of which show tissue specific distribution [4]. MH is expressed throughout the plant, with high amounts found in the leaves and seeds [5], whereas honokiol with its isomer magnolol are largely limited to the bark [6]. MH and 2-*O*-Methylhonokiol are isomers [7]. The optimal ratio of these two isomers and their mechanism of synthesis have not been fully characterized. Moreover, the effects of MH on the cardiovascular system remain poorly understood. Due to its low hydrophilicity, MH exhibits poor pharmacokinetics [8], which may lead to increased accumulation in the organs of the body. Some evidence has suggested beneficial effects of MH, as it has been associated with anti-inflammatory [9], anti-osteoclastogenic, anti-oxidative [10,11], and neuroprotective [12] effects. Considering the apparent non-specificity of these effects, it is likely that MH has low targeting efficacy and exerts its cellular protective properties through a wide range of mechanisms. In contrast to the observed beneficial effects, the co-treatment of compounds with MH counterparts, both magnolol and honokiol, can exert synergistic cytotoxicity [13]. The incongruity between the protective and detrimental effects of MH thus far in the literature highlights the importance of understanding how MH or other compounds extracted from magnolia affect the development and function of tissues and organs, like the cardiovascular system.

Both angiogenesis and vasculogenesis (de novo blood vessel formation from embryonic precursors) have many features in common, and impairment of these processes can in turn cause damage to organs and influence blood circulation. We aim to identify exposure windows, which can provide insight into the potentially toxic effects of MH on the development of the heart, angiogenesis, and vasculogenesis, beyond its other potential action on multiple sites through different toxicity pathways. Identifying the critical stage of MH-induced cardiovascular toxicity lays down a basis for further elucidation of an adverse outcome pathway for MH and provides a starting point for future studies on the mechanisms of MH toxicity.

Due to its inherent low concentration in plant extracts containing honokiol and magnolol, MH has not been fully characterized. MH, like propofol, has a phenol ring, which produces side effects causing hypertension and altering both heartbeat and heart rate. Since magnolia bark extract is gaining widespread popularity as a preventive and alternative to medical treatments [14,15], it is vital to understand the molecular mechanisms of MH and characterize whether it causes embryotoxic and teratogenic effects in the cardiovascular system in a variety of vertebrates. By accounting for heterogeneities typical of Japanese medaka (*Oryzias latipes*), which shares 58% homology with its human counterpart, our first vertebrate model for the embryonic lethality of MH enables us to investigate the effects of MH on the embryo’s developing heart and deformities, as well as alterations in inflammatory and parameters of coagulation.

In this study, we assessed the toxicity of MH in different life-stages of the medaka cardiovascular system. We hypothesized that the differential susceptibility of the stage-specific embryo might identify critical exposure windows to MH, which could in turn produce lethal and sublethal thresholds for toxicity because of differences in uptake of MH and subsequent internal concentrations.

## 2. Results

### 2.1. Toxicity of MH in Medaka Embryo

The toxicity of MH exposure on medaka was assessed at various stages of development (see MH-treatment in Materials and Methods section). While 10 μM MH was not toxic to embryos after 48 h, it caused 70% mortality within 96 h. A full concentration-response analysis of toxicity following six days of MH exposure time point revealed significant mortality of MH-treated medaka embryo/larvae (LC_50_ = 5.3 ± 0.1 μM) (Figure 1A). Larval toxicity was also evident, with the most common effects being spine malformations and edema. Observation of overall mortality with increasing concentrations over time revealed a reduction in larval survivability following exposure to concentrations of 2 μM MH and higher (Figure 1B). Some spinal deformities were observed in the larvae (Appendix B, Appendix A) which were associated with 5 μM and 10 μM MH. Embryos exposed to 10 μM MH showed delayed growth and high mortality rates during late larval- and juvenile-life stages.

A second batch of embryos exposed to MH later in development (2–6 dpf) also exhibited concentration-dependent toxicity (10 dpf, LC_50_ = 9.9 ± 0.2 μM) (Figure 1C). Like the first batch of embryos, survivability was significantly reduced in later-staged larvae exposed to 10 and 20 μM (Figure 1D). There was a direct relationship between the changes in MH concentration and changes in hatching efficiency. Behavioral alterations such as difficulty in swimming and equilibrium loss were seen with both 5 μM MH and 10 μM MH. However, while termination from MH treatment allowed 5 μM MH-treated embryos to recover normal behavior, stopping MH treatment at 6 dpf from hour 5 was ineffective in the 10 μM MH group. The control group treated with DMSO, ranging from 0.02 to 0.04%, exhibited normal embryo development and normal hatching. Embryo mortality for this group was always below 10%, contrasting clearly with the experimental groups. The sensitivity of adult medaka to MH toxicity was then assessed. Exposure of 5 μM MH in adults caused 65% mortality within 24 h, whereas both the control group and fish exposed to 1 μM MH survived (Figure 1E). The survival time of 5 μM MH was significantly less than that of 1 μM MH-immersed adult medaka. This result demonstrates that 5 μM MH reduces survivability in the adult medaka model and further underscores its importance for additional health-related research. 

The membrane penetration properties of MH on the medaka chorion are unknown. Although the chorion of the egg acts as a barrier within the embryo, the 80% mortality of embryos indicated an incorporation of MH into the egg. Mass spectrometry analysis of MH levels in conditioned growth media from the medaka revealed that the concentrations of MH decreased to non-detectable levels from time 0 h to 24 h (Figure 2A,B). However, a second molecule with a retention time of 2.628 min was present following 24 h incubation when MS was operated in scan mode (Figure 2C). Since the molecular and biochemical basis of formation of this molecule from MH is unknown, the identity of this unknown molecule was not characterized. Nevertheless, incubation of embryos with MH for 24 h led to 100% MH disappearance from the media (Figure 2D), suggesting either uptake of MH into the fish and/or degradation of the molecule. 

### 2.2. Exposure to MH during Early Development Affects Cardiovascular Structure and Function

Cardiovascular changes in response to MH were then assessed. We demonstrated here that MH treatment resulted in reduced blood flow which peaked at around an 80% decrease with 10 μM MH at the end of the 0–6 dpf treatment session (Figure 3A). The reductions in blood flow were associated with corresponding changes in blood vessel occlusion (Figure 3B). Reduced blood flow (Figure 3C) and blood vessel occlusion (Figure 3D) were noticeable in the 2–6 dpf embryos also, at the same concentrations as compared to those of control embryos from day six (stages 36–38). 

As shown in Figure 4A, the resting heart rate for both the control and DMSO-treated (0.02%) embryos increased with development. MH-treated embryos had lower heart rates than the control group at three-days and six-days post-treatment. MH treatment was effective in reducing the heart-beat of late embryos (6 dpf) (*p* ≤ 0.05, *n* = 27) by 9%, suggesting MH influences the normal functioning of the heart. High-speed time-lapse analysis of the heartbeat showed that bradycardia occurred in the 0–6 day embryos treated with 10 μM MH when treatments were initiated early (from hour 5) (Figure 4B,C; see videos for better visualization of the differences in heart malfunction between the control (video 1) and MH-treated embryo (video 2) in Appendix C under Appendix A). The average heart rate and blood flow for the 10 μM MH-treated embryos were lower than for the DMSO-treated controls. 

### 2.3. MH Is Implicated in Cardiovascular Dysfunction

To examine the mechanism of these vascular changes, expression levels of three key coagulation factors, factor XI (FXI, a protease of intrinsic pathway), factor VII (FVII, a protease of extrinsic pathway), and factor X (FX, a protease of common pathway) were quantified (Figure 5A). These factors are essential for hemostasis and indispensable for thrombosis. FXI, FX, and FVII circulate in the blood in zymogen forms. Activation of them leads to the formation of blood clots. As shown in Figure 5, these three protease transcripts were significantly (*p* ≤ 0.05, *n* = 100, replicated four times) elevated in embryos treated with 10 μM MH for six days from hour 5. 

Since MH reduced blood flow (Figure 3A,C), we hypothesized that the reduced blood flow activated the endothelium to synthesize and release tissue plasminogen activator (tPA, a fibrinolytic peptide) and plasminogen activator inhibitor 1 (PAI-1, a prothrombotic peptide) and thus decrease the ratio of tPA/PAI-1, ultimately promoting thrombosis. PAI-1 cDNA product was significantly elevated at the mRNA level with 10 μM MH, while tPA and endothelin B mRNA levels were not altered (Figure 5A). However, the ratio of the steady-state tPA/PAI-1 mRNA was significantly decreased in the embryos, suggesting increased levels of fibrin fragments, as previously described [16]. We measured the quantitative expression pattern of urokinase plasminogen activator (uPA) in 0–6 day embryos treated with 10 μM MH from hour 5. uPA mRNA levels were significantly increased (Figure 5A). To confirm that vascular endothelial cells are activated in response to MH, we examined the expression profiles of endothelin B receptors that are located primarily in vascular endothelial cells and angiotensin II type 1 receptor-associated protein (ATRAP), which is highly expressed in the kidney [17] and large vessel [18]. Vascular expression of ATRAP was significantly enhanced in response to MH (Figure 5A). Taken together, the data of Figure 5 demonstrated a previously unrecognized effect of MH on the control of the cardiovascular system and suggested that FXI, FX, FVII, PAI-1, uPA, and ATRAP are targets of MH. 

To further confirm that MH influences the normal functioning of the heart, we measured the expression patterns of heart development-related genes. Controls were exposed to 0.01% DMSO for the RT-qPCR study. After treatment with 10 μM MH, the embryos showed highly up-regulated expression of brain natriuretic peptide A and troponin T (Figure 5B). Further investigations were performed to explore the effect of MH on Nrg-2 and ErbB3, the molecules involved in the synthesis of acetylcholine at the neuromuscular junction [19]. Nrg-2, a cardiac chamber maturation marker, and ErbB3, which is involved in proper heart morphogenesis and function, were attenuated (Figure 5B). These findings conferred that the cardioprotective property of the Nrg molecule was compromised in the presence of MH.

### 2.4. MH Possesses the Proinflammatory and Pro-Oxidative Properties

We looked into the effect of MH on the expression levels of catalase, glutathione peroxidase (GPX), glutathione-S-transferase (GST), and superoxide dismutase (SOD) (Figure 6). Forkhead boxO1 (FoxO1) was overexpressed with 0–6 dpf MH exposure beginning from hour 5 (Figure 6). Catalase, GPX, and GST mRNAs were significantly overexpressed, whereas SOD mRNA was not statistically significant (Figure 6). This finding suggested that MH-dependent increased expression of these anti-oxidant enzymes in the embryo was a mechanism by which they could eliminate excess reactive oxygen species (ROS). Since the Wnt/β-catenin signaling pathway is capable of regulating inflammatory cell migration and macrophage phenotypes in zebrafish [20], we then assessed the expression profiles of tissue necrosis factor-alpha (TNF-α) and interleukin 1 beta (IL-1 β) (Figure 6). Expression analysis showed significant upregulation of IL-1β and TNF-α in MH-treated embryos at the transcription level (Figure 6). This observation suggests that the overexpression of FoxO1 might cause a decrease in endothelial cell sprouting and migration, as seen in a previous report [21]. On the basis of these results, MH appears to possess proinflammatory properties at the embryo level. 

### 2.5. MH Reduces the Normal Hatching Process of Medaka Embryos

We hypothesized that decreased heart rate might reduce the embryos’ activity, leading to a reduction in the distribution of nutrients and both coagulation and hatching enzymes, as has been previously suggested [22]. To test this hypothesis, we explored the effects of MH on hatching. MH caused a reduction in hatching in embryos exposed to 10 μM MH starting from the hours immediately after fertilization (Figure 7A) or starting two days later (Figure 7B). Our findings demonstrate that MH can influence both early and late developmental stages of medaka, leading to reduced hatchability of eggs at higher concentrations.

### 2.6. MH Regulates Wnt/β-Catenin Pathway during Cardiomyogenesis

Transcript levels of several Wnt/β-catenin signaling pathway proteins were altered in embryos treated with 10 μM MH for six days (Figure 8). Wnt 1 mRNA expression was significantly increased by MH. As shown in Figure 8, control and MH-treated embryos had similar levels of TGF-β2 mRNA expression, suggesting that MH could potentially promote cardiac fibrogenesis through the Wnt signaling pathway. 

High expression levels of Fzd 2 (Wnt 1 receptor), LRP5 (Wnt 1 co-receptor), and Dvl were observed in MH-treated embryos, but there was no difference in expression levels of glycogen synthase kinase-3 beta (GSK-3β), which is involved in the suppression of the Wnt/β-catenin and subsequent degradation of beta-catenin (Figure 8). A significant increase was apparent in levels of β-catenin transcript of the MH group. Increased β-catenin is implicated in ventricular myocyte proliferation control, while its decrease leads to differentiation [23]. Thus, MH might inhibit cardiac differentiation via increased endogenous β-catenin-mediated signaling during normal cardiac development.

Expression levels of dickkopf 1 (DKK1, a secreted Wnt/β-catenin pathway inhibitor) in the MH-treated embryos were significantly higher compared to those of the control group (Figure 8). MH-induced Wnt 1 overexpression enhanced the expression of several downstream molecules involved in heart development. These molecules included Fzd 2, Dvl, LRP5, β-catenin (a downstream target of LRP5), and DKK1. Collectively, MH clearly influenced the Wnt/β-catenin signaling pathway, which had roles during various stages of cardiac development. 

### 2.7. MH Prolongs Swimming Duration

The locomotion of embryos exposed to a sublethal concentration of MH was assessed two days post hatching (Figure 9). Control fish showed clear response to the light-dark cycle while MH treatment blunted this locomotor response (Figure 9). 

## 3. Discussion

Much of our knowledge of the function of MH has been extrapolated from its analogs’ action and its usage in the form of plant extracts in health and disease, which could provide a bias leading to untoward side-effects. Compounds/drugs often have diverse and/or mixed effects from one organ to another as well as across different disease states. The present study was aimed at demonstrating the effects of MH and to establish its possible therapeutic utility. Many plant extracts have MH-like molecules, but the effects of MH on cardiogenesis and neurogenesis have not been comprehensively studied in contrast to cancer and inflammation. With increasing interest in honokiol related compound therapy, and their usage as supplements in diets and cosmetics, investigations were performed to answer the following questions: (1) What was the effect of MH on known genes involved in cardiac development and function in medaka embryos? (2) What were the morphological and physiological effects of daily MH treatment in medaka embryos? (3) What was the effect on embryonic survival and hatching of subjecting medaka embryos at different stages of cardiovascular development to various concentrations of MH for different lengths of time? (4) Was MH a prothrombotic agent? (5) What was the impact of sublethal concentration of MH on locomotion? 

Fertilized eggs of medaka at two developmental stage windows (9–36 or 25–36) were exposed to increasing concentrations of MH in an embryo medium. Our data showed that MH-induced embryonic fatality was developmental-stage-specific. The embryos were more sensitive to MH at early stages of development (9–25) than in late stages of development. MH-treated embryos exhibited cardiovascular complications with a spectrum ranging from reduced blood flow to blood vessel occlusion, thrombus formation, and slow heart rate. 

In recent years, it has become apparent that the Wnt/β-catenin pathway (Wnt signaling pathway) is essential for the regulation of numerous genes in embryogenesis [24], adult cell biology [25], tissue homeostasis [26], and disease [27]. The canonical (known as the morphogen pathway) signaling of this pathway is capable of upregulating the expression of a wide range of genes, which have roles in giving rise to the majority of cardiomyocytes [28]. Targeted pathways related to Wnt include cardiac differentiation [29] and cardiac remodeling [30]. Increased activated Wnt signaling has been recognized as a major pathomechanism in heart and blood vessels [31]. Uncontrolled activation of the Wnt signaling pathway has been implicated in the pathogenesis of cardiovascular disease [32] and inflammation [33]. The Wnt/β-catenin pathway not only has a major role in cardiovascular development, but it has been also proven that a specific isoform of Wnt, Wnt-5A, functions in the process of neurogenesis and establishment of functional connectivity [34], suggesting its tissue specificity. To understand why chronic MH treatment may have undesirable effects in embryos, we investigated the effect of MH on the components of the Wnt signaling pathway and its downstream targeted molecules, which may offer novel mechanistic insights that could pave the way to enhancing its clinical utility. 

Due to difficulties in the pharmacological approach and the absence of antibodies, quantification of gene expression profiling of medaka cardiovascular tissues was the alternative approach to clarify the role(s) MH plays in embryogenesis, and to assess the clinical utility of MH. MH-treated embryos showed increased expression of the Wnt 1 gene (Figure 8). This deregulation of the Wnt signaling pathway prompted us to target key members of this pathway, which had exhibited distinct temporal and spatial profiles of expression during normal embryogenesis, post-surgery [35], and disease states [36]. It has been established that when both the Fzd receptor and LRP5/6 form a complex with Wnt ligands, the Wnt/β-catenin signaling pathway is activated [37]. Thus, we assessed the effect of MH on the expression profile of Fzd receptors and co-receptor LRP5. MH upregulated the expression levels of both Fzd and LRP5, suggesting Wnt/β-catenin signaling activation is sensitive to MH concentration. Most importantly, the transcript level of Fzd was positively correlated with the Wnt transcript expression level. Wnt-mediated Fzd/LRP5 stimulation leads to cytosolic β-catenin accumulation. However, its accumulation and signaling are tightly regulated via Wnt-dependent and Wnt-independent mechanisms [37]. Evidence shows that β-catenin signaling is essential for proper vascular formation and the development and functioning of the heart [38]. Next, investigations were performed to measure the expression pattern of β-catenin mRNA in MH-treated embryos. There was an increase in β-catenin mRNA. Since β-catenin has a rapid turnover [37], we explored the expression of GSK3β transcript. Evidence indicates that GSK3β activity is reduced/inactivated in the presence of Wnt signal [37]. As shown in Figure 8, GSK3β expression was unaltered, which is critical to the activation of β-catenin-mediated signaling. Moreover, our data suggested that MH had no direct effect on GSK3β gene expression and the unaltered level of GSK3β might be due to overexpression of Wnt levels, its binding to the complex of Fzd-LRP5, and stabilization of β-catenin [37]. Inactivation of GSK3β can result in the translocation of β-catenin to the nucleus and, subsequently, the induction of β-catenin-dependent downstream target genes. 

Urokinase plasminogen activator receptor (uPAR) [39] is one of these target genes downstream of β-catenin that has roles in both thrombosis and complement system. tPA and uPA are two plasminogen activators which are capable of catalyzing the activation of plasminogen. While uPA is required for the generation of plasmin activity in tissue undergoing pathological remodeling, tPA is associated with plasmin-induced activation of latent TGF-β in the vessel wall [40]. TGFβ2, a cytokine, is involved in vascular function, and mutations in TGF β2 are found to be implicated in cardiovascular diseases such as vascular complications and aortic disease [41]. The activity of uPA is regulated by its specific receptor, uPAR [42]. The mRNA of PAI-1 and uPA were enhanced by MH treatment in medaka embryos, whereas tPA mRNA steady-state levels were unaffected (Figure 5A). Similarly, TGFβ2 mRNA was unaffected by MH (Figure 8). This coincides well with the finding in the vascular wall [40] that there is a potential direct correlation between tPA expression and TGFβ2 gene expression and/or activation. The tPA/PAI-1 mRNA ratio was significantly decreased in the 10 μM MH group compared to the control. This suggests that MH appears to influence the fibrinolytic system during embryogenesis. In addition, Dvl mRNA was significantly elevated in MH-treated embryos. Overexpression of Dvl can inhibit the phosphorylation of β-catenin by GSK3β leading to β-catenin stabilization, and can consequently promote expression of the downstream targets. Overexpression of Dvl-1 in an atorvastatin-treated rat model of balloon-injured carotid artery has been shown to reverse the treatment effects of atorvastatin on vascular smooth muscle cells and collagen expression [43], suggesting the anti-restenosis action of Dvl. Collectively, these results also confirm that there is a positive correlation between β-catenin increases and the level of downstream target gene expression. 

While DKK-1 is a transcriptional target of the p53 tumor suppressor [44] and β-catenin [45], it plays a functionally redundant but protective role [46]. It is capable of suppressing the expression of Wnt target genes [47] during postnatal angiogenesis [48]. MH causes an overexpression of the DKK-1 gene in embryos. Our data has demonstrated that both the Wnt/β-catenin signaling and its modulator, DKK-1, are increased in MH-treated embryos, highlighting their imbalance pattern of expression. Evidence indicates that the overexpression of DKK-1 is markedly associated with reduced cell proliferation [45], endothelial dysfunction, and concomitant platelet activation [49]. There is a positive correlation between DKK-1 expression and the recovery period following acute myocardial infarction (MI) [50]. FVII, FX, and FXI were elevated in MH-treated embryos, suggesting their consumption due to the activation of platelets. Activation of platelets lead to the formation of thrombus and reduced blood flow, which was apparent in MH-treated embryos. It is well-established that reduced blood supply to the myocardial tissues can result in ischemia and subsequent MI. On the basis of plaque rupture, platelets are found to be the cellular source of DKK-1 in patients with acute ST segment-elevated MI [51]. In addition, our study also demonstrates that MH is capable of upregulating the gene expression of brain type natriuretic peptide (a secretory cardiac neurohormone, BNP) and Troponin T, which has cardiac specificity and is essential for cardiac contractility [52]. There is a positive correlation between elevated troponin levels and ST-segment elevation MI (STEMI) [52]. Elevated BNP levels are a strong predictive marker of heart failure [53]. It is plausible to suggest that overexpression of the components of the Wnt1/β-catenin dependent pathway by MH is due to heart failure and ischemic areas in the embryo heart during the wound healing process following acute MI, whereas MH-induced increased DKK-1 gene expression is mediated via platelet-induced endothelial activation. 

Although this is the first work in the literature to have explored the effects of MH on inflammatory cytokines in medaka, MH surprisingly caused robust increased levels of inflammatory markers such as TNF-α and IL-1β (Figure 6) in contrast to those of honokiol and magnolol. Our study did not address the differences we observed in the MH-mediated increased inflammatory mediators. We do not have a ready explanation for the discrepancy regarding the cited anti-inflammatory action of MH and that of our study. However, it is tempting to suggest that the elevation of these cytokines could be the result of at least two potential explanation: (1) elevated inflammatory mediators in medaka embryos were the result of genetic differences passed down from parents in our medaka colony or (2) the additive effect of MH on constitutive expression of the aforementioned cytokines in medaka as previously seen in mouse embryos [54]. The underlying mechanism(s) for the increase in these cytokines in MH-treated embryos remains uncertain. The increases in TNF-α and IL-1β are positively correlated with the presence and extent of cardiac biomarkers level (Figure 5). There is a wealth of knowledge about the association between heart failure and circulating inflammatory cytokines [55]. It is tempting to suggest that MH-induced inflammation might be the result of local tissue injury due to lack of oxygen/nutrients or systemic inflammasome activation. However, further investigation is needed to determine the potential causative role these inflammatory cytokines play in the progression of MH-induced cardiac injury. 

## 4. Materials and Methods

### 4.1. Medaka Maintenance and Breeding

All fish work was performed in compliance with animal ethics guidelines as given by the Institutional Animal Care and Use Committee (IACUC) at the University of Mississippi according to the Association for Assessment and Accreditation of Laboratory Animal Care International (AAALAC). Embryos were collected in the morning and maintained on a 14 h:10 h light:dark cycle. The medium or test solutions as well as plates were autoclaved or sterilized. All experiments with medaka embryos were conducted in an embryo medium (17 mM NaCl, 0.4 mM KCl, 0.36 mM CaCl_2_, 0.6 mM MgSO_4_, pH 7.4, and 0.0002% methylene blue) at 26 ± 1 °C.

Natural breeding or in vitro fertilization (known as squeezing) was used as a preferred method of choice for generating embryos for pharmacological treatment experiments. The latter techniques were used to generate embryos for most of our experiments, unless otherwise stated. 

### 4.2. Extraction and Isolation of MH

The seeds of *Magnolia grandiflora* were collected at the University of Mississippi campus (MS 38677). Air-dried powdered seeds (107 g) were soaked in EtOH (200 mL × 2 × 24 h each). The combined extract afforded oily material, where 2 g was subjected to centrifugal preparative thin layer chromatography (CPTLC, Chromatotron^®^, Analtech Inc., Newark, DE, USA), using a 6 mm silica gel rotor. The sample was dissolved in dichloromethane (DCM) and applied to the rotor under a rotation of 700 rpm, and subsequently eluted with *n*-hexane, then DCM, to end up with MeOH (200 mL each). Eighteen fractions were monitored and collected via TLC analysis (silica gel; solvents: *n*-hexane-EtOAc; 75:25). The fractions were visualized by spraying the TLC plates, with 1% vanillin-H_2_SO_4_ used as a spray reagent, where fractions 6 (361 mg) and 7 (36.0 mg) contained MH with a purity of 95% and 85%, respectively, via LC and NMR analyses. Further purification of fraction 6 was completed using a silica gel solid phase extraction cartridge (SPE) and was eluted with gradient of *n*-hexane/EtOAc (100:0→99:1) with 0.1% increments to afford 10 fractions. The fractions were monitored and pooled by TLC analysis (silica gel; solvents: n-Hex-EtOAc; 75:25), where fractions 7–9 afforded MH with a purity of 95% via LC and NMR analyses. LC analysis was conducted using an Agilent 1100 high performance liquid chromatography (HPLC) system equipped with a degasser (G1379A), quaternary pump (G13311A), auto sampler (G1313A), column oven (G1316A), and UV-Diode detector (G1315B) controlled by Chemstation software. Analysis of the fractions was carried out on an RP-C18 column (150 × 4.6 mm; particle size 5 µm; Luna) with column oven temperature set to 25 °C and a gradient system of eluent water (A) and acetonitrile (B) used. The gradient condition was as follows: 0–2 min (10% B), 2–30 min (10% B→90% B), 30–35 min (100% B). The flow rate of the solvent was 1.0 mL/min and the injection volume was 20 µL. All the analysis was carried out at wavelengths of 254, 280, and 325 nm with a run time of 35 min. HPLC-grade acetonitrile and water solvents were used. Acetic acid was added as a modifier to achieve a final concentration of 0.1% in each solvent. NMR spectra were acquired on a Varian Mercury 400 MHz spectrometer at 400 (^1^H) and 100 (^13^C) MHz in CDCl_3_, using the residual solvent as an internal standard. Multiplicity determinations (DEPT) and 2D-NMR spectra (HMQC, HMBC, and NOESY) were obtained using standard Bruker pulse programs.

### 4.3. MH Treatment

The up-and-down procedure (UDP) testing approach was used to determine the toxicity of MH on medaka, beginning at stage 9 (hour 5) of embryological development (as delineated by [56,57]) (Figure 10). For the analysis of the effects of pharmacological treatments on cardiac rate, thrombus generation, and blood vessel occlusion, fertilized eggs were collected within 5 h of mating and immersed in embryo medium containing various concentrations of MH (1, 2, 5, 10, and 20 μM) and 0.02–0.04% DMSO (control group) in 48-well culture plates. The first batch of embryos (0–6 dpf, *n* = 8 to 12 embryos/group) was exposed to five different concentrations (1, 2, 5, 10, and 20 μM) of MH and vehicle sample, 0.02–0.04% DMSO, which is known to be safe [58] and increase the permeability [59] of the embryo’s chorion. The second batch of embryos (2–6 dpf) was exposed to similar dilutions of MH (1, 2, 5, 10, and 20 μM) and 0.02–0.04% DMSO. All embryos were placed in a 48-well plate with 1 mL of their respective dilutions. Both batches were monitored for changes in morphology, delayed growth, behavior alteration, and mortality throughout the embryonic and larval-, juvenile-, and adult-life stages.

Embryos and larvae were raised in 48-well plates and maintained at a density of 1/1.0 mL in embryo medium with daily medium change. Groups of medaka embryos, often numbering 8 to 12, were exposed for an exposure period from 0 to 6 days post fertilization (organogenesis period at stages 9 to 36–38), and 2 to 6 dpf (stages 25 to 36–38), with an interval of at least 24 h. Embryos/hatched fry were reared in normal embryo medium at 6–10 dpf without MH (washed groups). Survivability, hatching efficiency, blood vessel occlusion, and blood flow frequency were observed until 10 dpf. Heart beats were counted on 3 and 6 dpf, and RNA isolation was immediately performed after 0–6 dpf exposure. Behavioral experiments were conducted two days post hatching (dph) after 0–6 dpf MH exposure. Larvae with no obvious malformation were used for locomotion experiments. Both embryos and larvae were monitored daily by imaging.

### 4.4. Determination of MH Purity and MH Absorption in Medaka Embryos

The absorption of MH from embryo medium to embryos were ascertained after a single concentration of MH (10 μM) for a period of 24 h using an Agilent 1290 Infinity series UHPLC with a diode array detector and an Agilent 6120 quadrupole mass spectrometer (Agilent Technologies, Santa Clara, CA, USA). The highly purified MH reference standard showed a base peak of [M + H]^+^ 281 with the APCI positive ionization mode; this ion was thus used in a selected ion monitoring mode (SIM) to detect MH in the samples. Embryos were exposed to embryo medium containing 10 μM MH from 5 h post fertilization (hpf) to 24 hpf. The disappearance of MH in the embryo’s bathing (conditioned) medium was regarded as the absorption characterization of MH in the embryo toxicity test and the cardiovascular toxicity test. The identity and concentration of MH in conditioned medium was measured by UHPLC/MS. 

The toxicity testing involved multi-stage exposure with repeated MH concentrations for six consecutive days with an interval of at least 24 h. Six groups of embryos (*n* = 12/group) were treated concurrently with 1, 2, 5, 10, and 20 μM MH and 0.02–0.04% DMSO (control group) at 5 hpf. There was a washout period between each treatment to clear any remaining free MH and extruded molecules into the conditioned embryo medium. The stock solution of MH was prepared in 100% DMSO (Sigma-Aldrich, St. Louis, MO, USA). Treatment was done according to the design depicted in Figure 10. In another set of experiments, hatched fry were reared in normal embryo medium without MH until 10 dpf (washed groups). 

### 4.5. Microscopy Study

Control and treated fish were sampled at various stages of development according to the treatment schedule. Cardiovascular structure, blood flow, and heartbeat were analyzed under a microscope. Images were acquired on a Nikon TI2-E inverted microscope with a white light LED illuminator and a sCMOS Cooled Monochrome Camera. Videos were captured using Nikon Elements automated acquisition at a rate of 20 frames per second for one minute.

### 4.6. Locomotion Study

The free-swimming behavior of MH-treated larvae was compared to that of the control group. MH was prepared at a 5 μM concentration in embryo medium. This concentration was selected on the basis of LC_50_ studies. The effects of acute MH exposure on locomotor activity in larvae were examined at 2 dph. We assessed activity in 24-well plates. Larvae were placed individually in wells containing 2 mL of embryo medium. The larvae acclimatized to the darkness of the Zebrabox over 20 min (Viewpoint, Montreal, Canada) before the start of the locomotion experiment. The duration of movements was measured at a velocity of ≥2 mm/s.

### 4.7. RNA Extraction

The isolated heart of the medaka embryo from the 6 dpf control and MH-treated embryos was too small to obtain from it enough total RNA for the quantification of target genes. To overcome this problem, 100 embryos from the 6 dpf control and MH-treated embryos were pooled into one sample and RNA was extracted using an RNAeasy micro kit (QIAGEN GmbH) following the manufacturer’s protocol. However, samples were replicated four times.

### 4.8. Reverse Transcription—Quantitative Polymerase Chain Reaction (RT-qPCR)

One microgram of total RNA extracted from each sample were reverse-transcribed into cDNA using a Quantitect RT kit (QIAGEN GmbH). Quantitative expression profiles of the genes of interest were analyzed using SYBR Green (Invitrogen, Carlsbad, CA, USA) according to the instructions of the manufacturer. Primers used for the amplification of each gene have been tabulated in Table 1. 

Quantification of target genes were done using an ABI 7000 real-time PCR machine (Applied Biosystems, Inc., Foster City, CA, USA). To measure the relative quantity of target genes per 1 μg of the total RNA from each group, a 2^−∆∆*C*t^ method was used. On the basis of McCurley and Callard’s 2008 study [60] as well as our own observations, eEf1α was included as an internal control. Samples were replicated four times. Quantitative expression data were used as the basis for making the major interpretations of this study. 

### 4.9. Statistical Analyses

Statistical analysis was performed using Graph Pad Prism V6.0. Data are presented as mean ± SEM. Morphological data were analyzed by one-way ANOVA followed by Tukey’s post-hoc multiple comparison test where more than two groups were compared. The LC_50_ was calculated by log transformed data using nonlinear regression (curve-fit) (GraphPad Prism). Statistical analysis for all RT-qPCR data was performed by two-way ANOVA followed by post-hoc Bonferroni test. Data for locomotion study were analyzed by two-tailed *t*-test. A difference between two means was considered to be significant when *p* < 0.05 (* *p* < 0.05). 

## 5. Conclusions

Natural product extracts play complex roles in cardiovascular homeostasis. They can have protective and exacerbating effects on diseases due to the presence and complexity of plant extract composition. There has been very little information produced about the pharmacodynamics and pharmacokinetics [61] of MH, particularly in regard to both its teratogenic and cardiovascular effects. The typical recommended levels of magnolia bark extracts range from 200–800 mg/day/person [62]. The potential effects of MH on embryonic development are prominently apparent in our current study (Figure 11). Since medaka can be used to model the human cardiovascular system [57], here we report that MH alone is harmful to embryos because of its proinflammatory and prothrombotic properties as well as its effects on the Wnt signaling pathway. Unfortunately, little is known about the contribution(s) of MH once the trajectory has been set following its ingestion in humans. Our evidence is not sufficiently robust to support and extrapolate its deleterious effects in humans. However, we suggest that its inclusion in plant extracts could potentially retard the beneficial effects of other components of magnolia bark extracts or others. Herbal medicine optimization research must take MH levels into consideration in order to prevent the stimulation of both stress-induced pathways and the Wnt signaling pathway, which plays major roles in the control of all facets of embryonic development. Complementary approaches are needed to have a better understanding of the effects of maternal use of plant extracts containing MH on the offspring’s health during pregnancy.

## Figures and Tables

**Figure 1 molecules-24-00475-f001:**
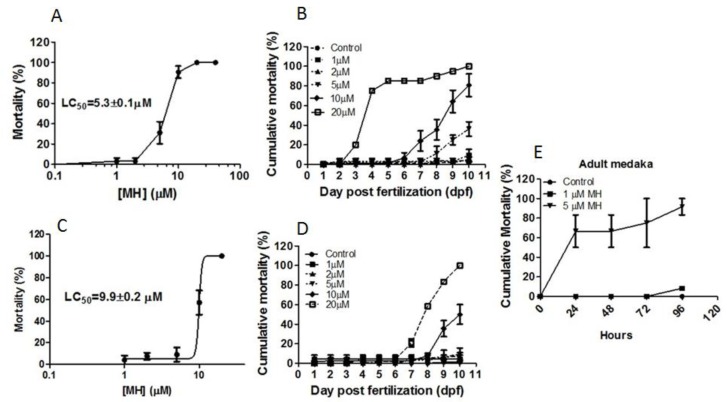
Cumulative mortality of medaka embryos. Embryos were exposed to 1, 2, 5, 10, and 20 μM 4-*O*-Methylhonokiol (MH) for 0–6 days post fertilization (dpf) (**A**,**B**) or 2–6 dpf (**C**,**D**). (**A**) Embryos were treated with MH for six days from hour 5, and the LC_50_ was calculated based on mortality observed at 10 dpf. (**B**) Cumulative mortality of all embryos until 10 dpf treated with various concentrations of MH for 0–6 dpf. (**C**) Embryos were treated with MH for four days from 2 dpf and the LC_50_ was calculated based on mortality observed at 10 dpf. (**D**) Cumulative mortality until 10 dpf of all embryos treated with various concentrations of MH for 2–6 dpf. The LC_50_ was calculated by log transformed data using nonlinear regression (curve-fit) (GraphPad Prism). Each value represents mean ± SEM (*n* = 12, replicated five times). (**E**) The relationship between MH concentration and mean adult medaka mortality. Eighteen male medakas were randomly divided into three groups (*n* = 6). They were exposed to 1 or 5 μM MH, and monitored for 96 h. Controls were exposed to 0.02% DMSO.

**Figure 2 molecules-24-00475-f002:**
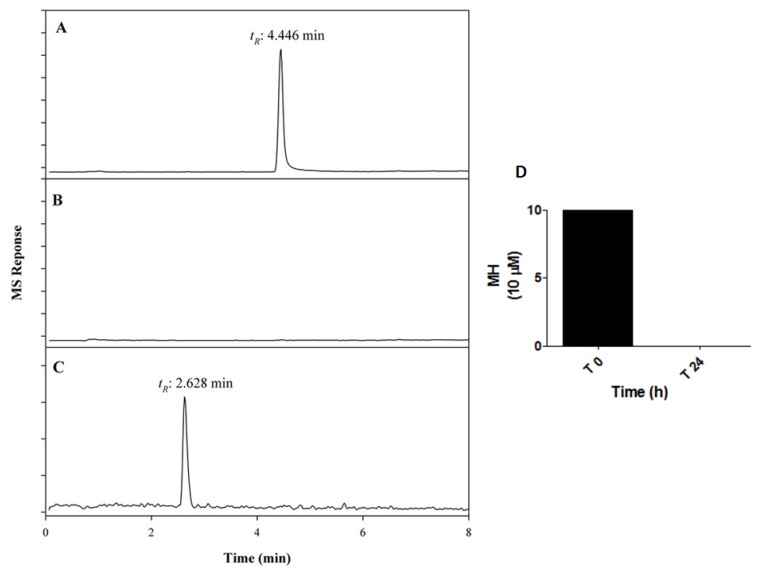
Quantification of MH uptake by medaka embryos by liquid chromatography-mass spectrometry (MS). This figure is a typical MS profile for the analysis of 10 μM MH uptake following 24 h-incubation with embryos, and maintained at 26 ± 1 °C. The amount of MH disappearance from the conditioned media was measured at time 0 h and time 24 h. Selected ion monitoring (SIM) at *m*/*z* 281 shows signals for MH at time 0 h (**A**) and time 24 h (**B**). (**C**) The unknown molecule in MS scan mode. (**D**) The amount of disappearance of MH following 24 h incubation with embryo.

**Figure 3 molecules-24-00475-f003:**
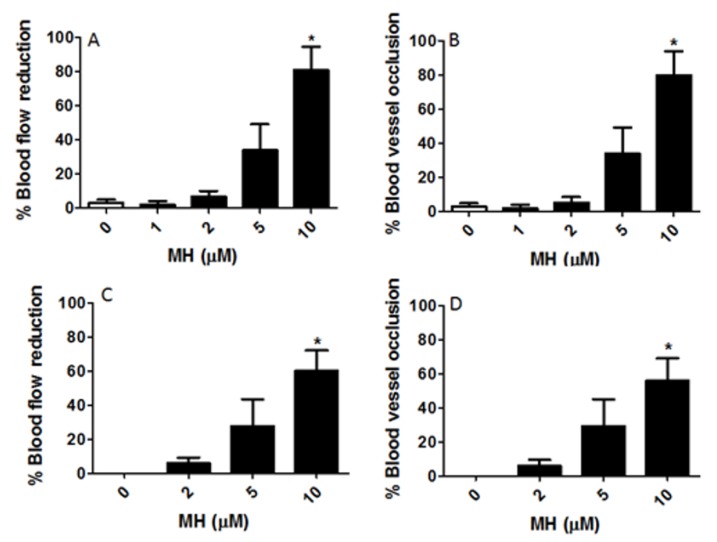
Reduction of blood flow and vascular occlusion by MH. Embryos were exposed to 1, 2, 5, and 10 μM MH from 5 h post fertilization (hpf) to 6 dpf (0–6 dpf, (**A**,**B**)) or from 2 dpf to 6 dpf (2–6 dpf, (**C**,**D**)). This figure showed that the reduction in blood flow was proportionally correlated with blood vessel occlusion. (**A**) Blood flow was observed on days 2, 3, 4, 5, and 6 to verify the duration of the reduction of blood flow. This panel exemplifies the percent reduction in blood flow until 10 dpf in response to various concentrations of MH for 0-6 dpf. (**B**) This panel shows percent occlusion following MH treatment. During the treatment, blood flow was significantly reduced compared to the control immediately after occlusion. (**C**) Thirty-two embryos (2 dpf) were randomly divided into four groups (*n* = 8). Embryos were treated with the indicated concentrations of MH and 0.02% DMSO (control) in embryo medium and were maintained at 26 ± 1 °C for 4 days. This figure shows reduced blood flow in MH-treated embryos (2–6 dpf). (**D**) This figure demonstrates a blockage that prevents normal flow of blood for the MH-treated embryos exposed to various concentrations of MH from 2 dpf to 6 dpf. Each bar represents data pooled from 4–5 independent experiments. Statistical analysis was performed by one-way analysis of variance (ANOVA) followed by Tukey’s post-hoc multiple comparison test. *p* < 0.05 was considered as significant. The asterisk (*) indicates values significantly different from the control.

**Figure 4 molecules-24-00475-f004:**
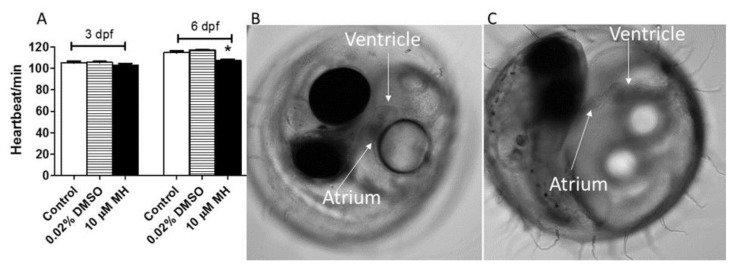
MH is responsible for the defect in cardiac function in vivo. Embryos were exposed to 10 μM MH or 0.02% DMSO (which was used as solvent in the treated condition) as the control from 5 hpf to 6 dpf. Heartbeat, blood flow, and heart structure were evaluated. (**A**) Heart beat was recorded and expressed as beats per min. Bar graphs of results obtained by counting heart beats. Results are given as the mean percentage of heart beat ± SEM (*n* > 8 embryos for each condition). Statistical analysis was performed by one-way ANOVA followed by Tukey’s post-hoc multiple comparison test (*p* < 0.05). (**B**) Representative image of heart function in the control. (**C**) Representative bright field images of MH-induced heart ventricle malfunction in embryos treated with 10 μM MH for six days from hour 5. See videos for better visualization of the differences in heart malfunction between the control (video 1) and MH-treated embryo (video 2). Images were acquired on a Nikon TI2-E inverted microscope with a white light LED illuminator and a sCMOS Cooled Monochrome Camera. Videos were captured using a Nikon Elements automated acquisition device.

**Figure 5 molecules-24-00475-f005:**
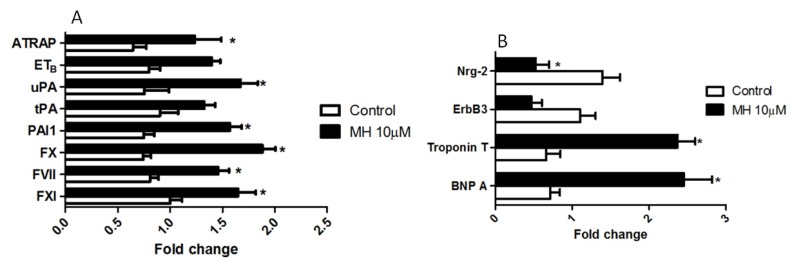
Embryos treated with MH lead to altered cardiac biomarker gene expression and altered expression of genes involved in thrombosis, fibrinolysis, and vascular tone. Real-time RT-qPCR was performed on total RNA isolated from each group of embryos using described primers in Table 1 leading to the amplification of the target gene. (**A**) RT-qPCR analysis of factor XI (FXI), factor VII (FVII), factor X (FX), plasminogen activator inhibitor 1 (PAI-1), tissue plasminogen activator (tPA), urokinase plasminogen activator (uPA), endothelin B (ETB), and angiotensin type 1 receptor associated protein (ATRAP) isolated from control and MH-treated embryos. (**B**) RT-qPCR analysis of natriuretic peptide A, Troponin T, ErbB3, and Nrg2. Data were normalized to the eukaryotic elongation factor 1-alpha (eEf1α) polymerase chain reaction signal. Each value represents mean ± SEM (*n* = 100, replicated four times). * indicates a value is significant versus the respective control group (*p* < 0.05).

**Figure 6 molecules-24-00475-f006:**
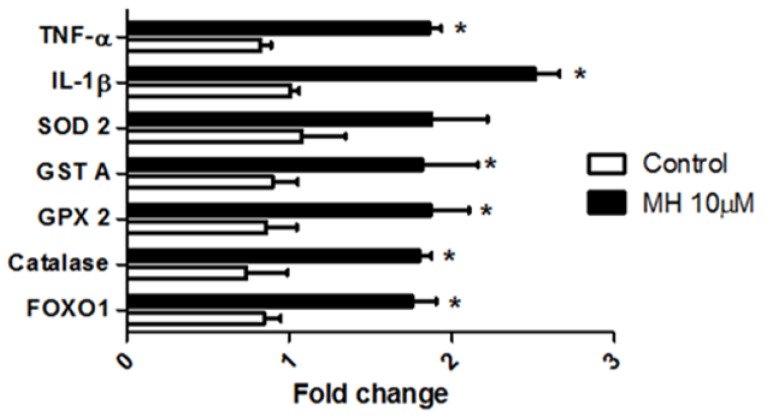
Embryos treated with MH lead to altered expression of genes involved in cell regulation, oxidative stress, and inflammation. Real-time RT-qPCR was performed on total RNA isolated from each group of embryos using described primers in Table 1 leading to the amplification of the target gene. RT-qPCR analysis of FoxO1, catalase, glutathione peroxidase (GPX2), glutathione-s-transferase (GSTA), superoxide dismutase 2 (SOD-2), Interleukin 1 beta (IL-1β), and tissue necrosis factor-alpha (TNF-α) isolated from control and MH-treated embryos. Data were normalized to the eEf1α polymerase chain reaction signal. Each value represents mean ± SEM (*n* = 100, replicated four times). Statistical analysis was performed by two-way ANOVA followed by post-hoc Bonferroni test. * indicates values which are significant versus the respective control group (*p* < 0.05).

**Figure 7 molecules-24-00475-f007:**
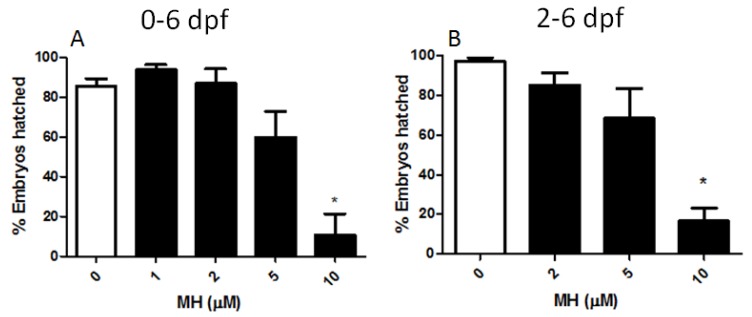
The effectiveness of MH at different concentrations at altering hatching of medaka embryos. Under pure culture conditions, embryo hatching efficacy was altered by increasing concentrations of MH. (**A**) Embryos were treated with increasing concentrations of MH for six days from hour 5 (0–6 dpf). MH concentrations of 5 μM and higher were required to cause a delay in hatching. Hatchability of embryos from six days MH (10 μM)-treatment was significantly decreased in comparison to untreated controls until 10 dpf. The external differences in appearance of the newly hatched control frys on day ten and the 10 μM MH-treated embryos were significantly obvious. (**B**) Embryos were treated with increasing concentrations of MH for four days from stage 25 (2–6 dpf). MH concentrations of 10 μM and higher were required to significantly cause a delay in hatching. Results are given as the mean percentage of hatching efficiency ± SEM (*n* > 8 embryos for each condition). Statistical analysis was performed by one-way ANOVA followed by Tukey’s post-hoc multiple comparison test. *p* < 0.05 was considered as significant. The asterisk (*) indicates values significantly different from the control.

**Figure 8 molecules-24-00475-f008:**
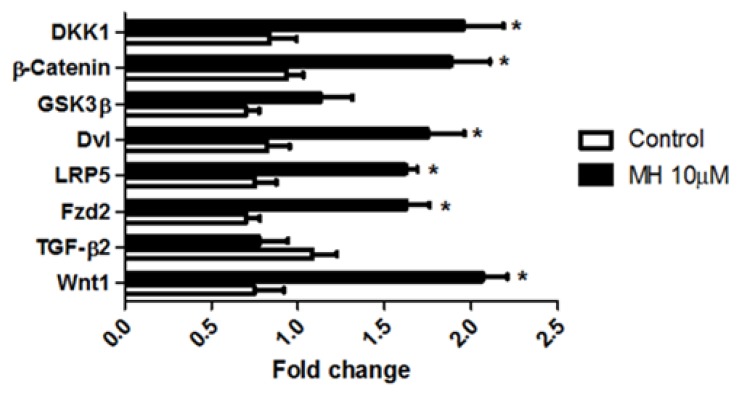
MH controls the Wnt signal transduction pathway, a main regulator of development. Real-time RT-qPCR was performed on total RNA isolated from each group of embryos using described primers in Table 1, leading to the amplification of the target gene. RT-qPCR analysis of Wnt 1, transforming growth factor beta 2 (TGF-β2), frizzled 2 (Fzd2), low-density lipoprotein receptor-related protein 5 (LRP5), dishevelled (Dvl), glycogen synthase kinase 3 beta (GSK-3β), β-catenin, and dickkopf 1 (DKK1) isolated from the control and MH-treated embryos. Each value represents mean ± SEM (*n* = 100, replicated four times). Statistical analysis was performed by two-way ANOVA followed by post-hoc Bonferroni test. * indicates values which are significant versus the respective control group (*p* < 0.05).

**Figure 9 molecules-24-00475-f009:**
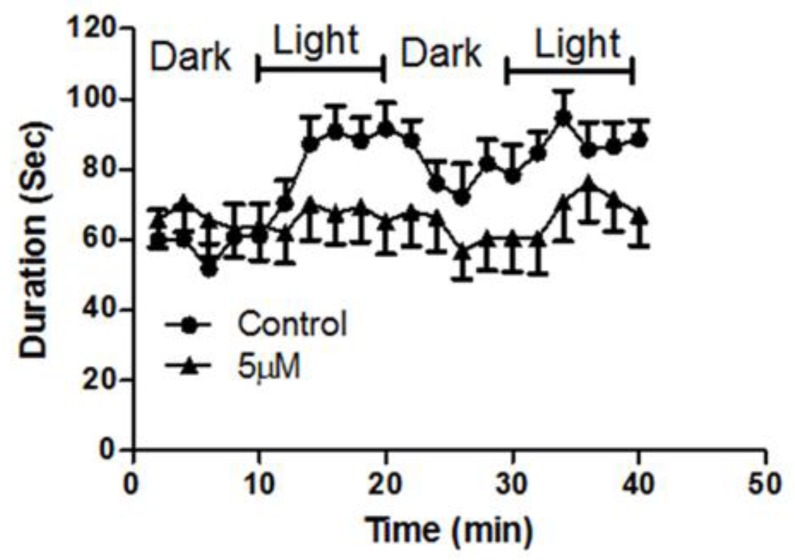
MH altered larvae’ locomotor responses to light and dark stimuli. Medaka larvae (2 days post hatching (dph)) were monitored for 40 min (0–10 min dark, 10–20 min light, 20–30 min dark, 30–40 min light) using a ViewPoint Zebrabox. Duration of movements was measured at 2 min intervals at a velocity of ≥2 mm/s. Larvae were treated with 5 μM, a sub-lethal concentration; MH and their activity were measured in 10 min windows at the indicated time points. No significant (two-tailed *t*-test, *p* < 0.05) increase in activity in MH-treated larvae (solid closed triangles, *n* = 14) was noted for the time window during the light-cycle compared to baseline levels and that of the control group (solid closed circles, *n* = 16). Each datum represents mean ± SEM of 16 observations.

**Figure 10 molecules-24-00475-f010:**
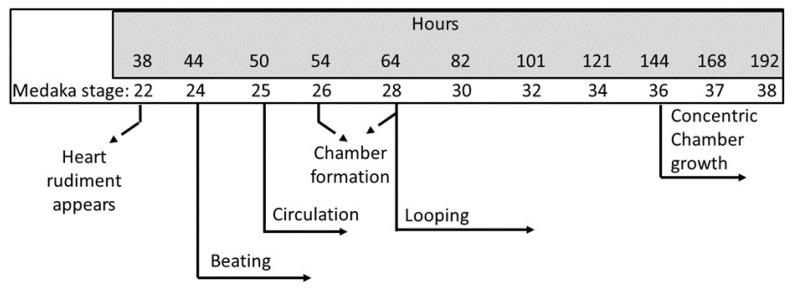
Stages with regard to hours post fertilization of the medaka heart. This scheme outlines the embryological development of the heart during the first 144 h and subsequent formation of concentric chamber growth. The medaka heart begins beating and pumping around 44 (stage 24) to 50 (stage 25) hours post fertilization. Medaka embryos were treated with MH either from 0–6 days post fertilization or 2–6 dpf at 26 ± 1 °C.

**Figure 11 molecules-24-00475-f011:**
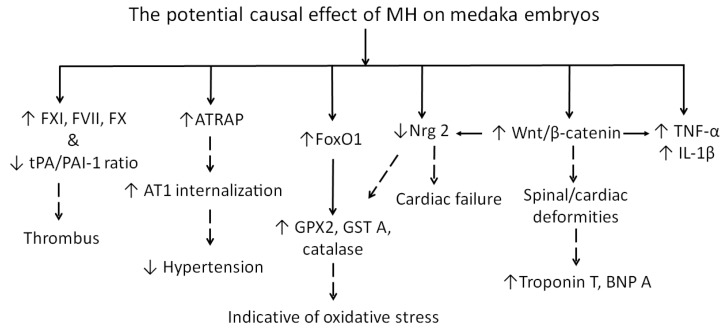
Flowchart depicting MH treatment induces cellular and morphological changes in medaka embryos that lead to inflammation, thrombosis, spinal, and cardiac deformities. Solid arrows indicate the direction of the effects of MH on indicators of increased inflammation/cardiovascular risk and the anatomic locations as well as a link with the prior object/event. The flow of physiological changes that require further investigations are shown in dashed arrows. Legend: ATRAP, angiotensin receptor-associated protein; AT1, angiotensin type I receptor; β-catenin, beta catenin; BNPA, brain natriuretic peptide type A; FVII, Factor VII; FX, Factor X; FXI, Factor XI; FoxO1, Forkhead box O1; GPX2, glutathione peroxidase 2; GST A, glutathione S-transferase A; IL-1β, interleukin-1 beta; Nrg 2, neuregulin 2; tPA, urokinase plasminogen activator; PAI-1, plasminogen activator inhibitor 1; Wnt, wingless/integrated oncogenes.

**Table 1 molecules-24-00475-t001:** Oligonucleotide sequences used in this study.

Gene	Sense (5′-3′)	Antisense (3′-5′)	Product Size (bp)	Reference
**eEf1α**	GGAGGCCAGCGACAAGAT	GCGAGAAGGTGGCAGGAT	115	NM_001104662.1
**FX**	TGTCAAAGCCCTGTGTGAAT	AGAAATGTTCACAGCCACCA	147	XM_020710825.1
**FXI**	GAAGGATAATGCAGACCAGTGTC	GATGACACCCTTCAAGTAGCATC	127	XM_004074394.4
**FVII**	GTTCTGTCGGATAGGTGGATTT	CCTCCAGGTCATGTTTACCTAC	97	XM_004066449.3
**PAI-1**	ATGCCGAGGTTTTCTCTGAAC	GTTGAACATGTCTCCCAGTCC	78	XM_020711407.1
**uPA**	ACTGTGTTTCTGGGAAAGAGTG	GGATGATCATTTTCTCCACGGT	82	XM_004077409.4
**tPA**	CAGCCCCGATCCAAGC	CCCTTCCATCGCAGCC	185	XM_011478844.3
**ATRAP**	CATGTGGGGGAACTTCAGC	GCCCACCAGAAACATGAGG	91	XM_011475678.2
**ETB**	CTGATCTTTGTGGTGGGCAT	CCCATTCCTCATGCACTTGT	78	NM_001104844.1
**IL-1β**	CTGTTTCTGGAGGAGGTGG	AGAAGAGGAAGCGCACATT	79	XM_011478737.2
**TNF-α**	AACCGAAGAGTCTGAGAGGG	AGCTGAAGAAGAGTACCGCT	105	XM_004074335.3
**Catalase**	TGCTAGCAGTTGATTGTCTGT	CACAGATCCACTGAAACAGGA	100	XM_004069460.2
**GPX2**	TCAACGGAGTAAACACGCAT	GATCCTGCATGAGAGAGCTG	90	XM_004082594.3
**GST A**	CTGAAGGAGAGCGGCAC	CAGGAACGAGCCAGAGC	107	XM_020710769.1
**SOD2**	AAATGTGCGTCCTGACTATGT	TTTTGGCTATCTGAAGACGCT	83	XM_004083471.3
**Wnt1**	CCAGAAAACCCAGCTCACAA	TTGTGGGAGCAGAAGTTTGG	80	XM_020704658.1
**FZd2**	CACATGACCCCAGACTTCAC	AGAAACCAGAAGTGATGCCG	76	XM_020705151.1
**LRP5**	GAAGGCCCGAGCAGTTCA	AAGACATGGCTCCGTCGT	101	XM_011472833.2
**Dvl**	TGCTGAAACAAAGCCCAAAGT	ACCTCAAGGATCTGAGTGAGC	87	XM_011490628.3
**β-catenin**	CACAGAACTCCTACACAGCC	AGGCGCTTCTTGTAGTCTTG	102	XM_004077778.3
**DKK1**	GTGACACATGCCTGAGATCG	CACAGGCTTACAGATGCGAG	83	XM_020709512.1
**GSK3β**	AGCTGCAGATTATGAGGAAGTTG	TAGACGGTCTCTGGAACATAGTC	130	XM_023950884.1
**ErbB3**	GAGGTTGAGAAGGATGGCGT	CTACCTGGACTTCCTGTGCC	86	XM_011474665.3
**NRG-2**	CTCGTCACTGTGGGGGA	CTCGTCAGTGGGGTCCA	93	XM_020708867.2
**BNP A**	GAGCTCTGTTGATGAGGAGG	CAGTCCTGGCTCATCTTCTC	88	NM_001104685.2
**Troponin T**	GAGCTCTGTTGATGAGGAGG	GCTGATCCGGTTTCTGAGT	362	XM_004068713.4
**FoxO1**	GCCCATGCCAGTTCTGAGTA	ATCCTCCGTGTTGGTGGATG	102	XM_011485361.2
**TGF-β2**	GTTACTCCGACCTGAGGAAGATAG	TGACACCCAATCTTTAACGGTTTC	127	XM_004073149.3

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
