# Peer review of "4-O-Methylhonokiol Influences Normal Cardiovascular Development in Medaka Embryo"

_molecules, 2019, doi:10.3390/molecules24030475_

Round 1
Reviewer 1 Report
Letter to Authors
molecules-417817
4-O-Methylhonokiol influences normal cardiovascular development in medaka embryo
Santu K. Singha, Ilias Muhammad, Mohamed Ali Ibrahim, Mei Wang, Nicole M. Ashpole, Zia Shariat-Madar
181219
Dear authors,
This MS is an interesting and may be important report dealing with negative aspects of 4-O-Methylhonokiol, a major component of magnolia bark extracts, to embryogenesis of a model vertebrate, medaka. Since magnolia bark extract is used as a herbal medicine or diet supplement, and since positive effects of 4-O-Methylhonokiol are recently reported, publication of this MS telling possible unwanted side effects of this herbal compound poses a question on medicinal or supplemental use of this herbal compound to be controlled. It is thus worth publishing of this MS in your journal, if adequately revised. See below for detail.
L9
No one is affiliated with the National Research Center of Egypt.
L27
Consider adding wards related to Chinese herbal medicine to attract wider readership, though I am not sure if there is a limitation of number of keywords.
L30
Evidence strongly indicates that (verbose) -> delete
L32
4-O-Methylhonokiol (MH) -> MH (4-O-Methylhonokiol)
Do not begin a sentence with Arabic numbers. Make a trick to avoid it.
L34
Magnolias -> magnolias
L35,48
Magnolia -> {in Italics, or in lower case}
Make sure whether this word is a Latin name or a common name in English. See also L60.
L42
This 42 suggests that -> delete
Words '.. suggest .. may' make this sentence dubious.
L50
refers to -> delete
L51
in common and -> in common, and
Insert a comma.
L67
coagulations parameters -> {coagulation parameters, parameters of coagulations}
L68 results
This section is a mess. Method-describing statements frequently interrupt this section. Tell your results straightforward, deleting long interrupting sentences for readers to see your results quickly. I suggest you to consult with the editor for re-organization of your MS into a classical paper structure (introduction - M&M - results - discussion). This is an easy way to clear your result section. If not allowed, write a draft once of the classical structure and then reorder sections simply.
L69-73
We assessed .. internal concentrations. -> move to the introduction section (L67)
L75-76
To date, .. development in medaka. -> move to the introduction section
L76-78
In this study, .. from hour 5 in development (stage 9). -> move to the M&M section or delete
L79 Fig.1
Replace the bit-mapped figure graphics with a vector data (eps format is recommended). Or otherwise, replace with a graphics with higher resolution (>600 dpi).
This figure is describing your experimental design and suitable for the method section.
L84-93
To determine .. stages. -> move to the M&M section or delete
L93
Here, we report that (wordy) -> delete
L98
I see mortality raise also in 5 uM. You should present the level of significance (p-value) and comparison framework (pairwise or among groups?).
L111-115
To compare the sensitivity .. at 28 оC for 96 h. -> move to the M&M section or delete
L115
Exposure of 5 uM MH -> Exposure of 5 uM MH to adults
Deletion of prior sentences needs some revision to fit in.
L121 Fig.2
Replace the bit-mapped figure graphics with a vector data (eps format is recommended). Or otherwise, replace with a graphics with higher resolution (>600 dpi).
L132
not known -> unknown
L133
percent -> %
L134-139
To assess .. concentrations. -> move to the M&M section or delete
Some revision of the next sentence may accordingly be necessary to fit with.
L156-157
First, .. cardiovascular damage. -> move to the M&M section or delete
L157
This study demonstrated -> We demonstrate here
L162
Justify this line fully both sides.
L181
a lower heart rates -> lower heart rates
L203-209
Investigations .. blood clot. move to the M&M or discussion section or delete
L209
these three protease -> factor XI (FXI, a protease of intrinsic pathway), factor VII (FVII, a protease of 205 extrinsic pathway), and factor X (FX, a protease of common pathway) coagulation factors
L218-258
Some parts should be moved to M&M section, while some others to discussion section.
L223
the fact that endothelium is activated -> possible endothelium activation
You hypothesized it. See L218. It is not a fact.
L236
Further experiments .. early warning signs. -> delete
Do not excuse here.
L238
Replace the bit-mapped figure graphics with a vector data (eps format is recommended). Or otherwise, replace with a graphics with higher resolution (>600 dpi).
Box-frame overlaps onto panel A.
L260-262
Next,.. SOD (Fig. 7). -> move to the M&M section or delete
L261
FOXO -> forkhead box-containing protein O (FOXO)
L275
unaffected?
The difference in expression levels is not significant, but ...
L277,425,457
Evidence indicates that (verbose) -> delete
L278-280
To learn .. (IL-1beta) (Fig. 7). -> delete or move to M&M section
L280
these two cytokines -> two cytokines, tissue necrosis factor-279 alpha (TNF-alpha) and interleukin beta (IL-1beta) (in Greek)
L285-286
However, further investigations .. via MH. -> delete
Do not excuse here. Do it in discussion section.
L294,303
What does '10 dpf' mean?
L350 says hatching typically takes seven days. Does hatching of MH-treated embryos retard three days?
L304
significantly obvious -> obvious
L312-314
Although several transcription factors .. with cardiac dysfunction. -> move to M&M or introduction section, or delete
L314-316
We analyzed .. and found that -> delete
L316
their transcript levels -> transcript levels of some key genes in the Wnt/beta-catenin (in Greek) signaling pathway
L317-319
Since both .. embryos as well. -> move to M&M or introduction section, or delete
L317
TGF-beta -> transforming growth factor-beta (in Greek)
L332-333
Next, .. pathway. -> delete
L336
beta -> in Greek
L341
Further studies determined that -> delete
L348-352
Since the fry stages of fish .. at a velocity of >_2 mm/sec. -> move to M&M section or delete
L353
circadian activity -> response to a light-dark cycle
The word 'circadian' means near-daily.
L355-360
The lack of clear locomotor activity .. is out of scope of this report.
You may simply say that MH-treated larvae were inactive in comparison with control group, instead of long unclear sentences.
L362
MH -> MH-treated
L369
solid close -> solid
L374
usage in -> usage of
L376
an effort -> aimed
L380
usage as supplements in diets and cosmetics
You should mention here or in the conclusion section about routine dose to diet supplements or to herbal medicine in comparison with your experimental dosage.
L445
In addition, Dvl mRNA was significantly elevated in MH-treated embryos. (redundant repetition of the results) -> delete
This sentence is well summarized in the next sentence as 'overexpression of Dv1'.
L468
According to the American College of Cardiology, -> give a reference instead
L503
MH -> of MH
L506
gm?
L508
dichloromethane -> dichloromethane (DCM)?
L688 references
Check carefully the list again from the beginning. Reference list is frequently a mess of errors. Do you follow the journal style? Is numbering and citation in the text consistent? Are author names and their spelling correct? Are journal names correct? Are publication years correct? Is title case for paper titles allowed? Is lower case for journal or book titles allowed?
L693
: RTP -> delete
L730
: JAD -> delete
L733
: an international journal published for the British Industrial Biological Research Association -> delete
L744
: an official publication of the American Association of Anatomists -> delete
L751
Aquat Toxicol -> Aquatic Toxicology?
L810
Circulation. Heart failure -> Circulation: Heart Failure
L834
: official journal of the Histochemistry Society -> delete
L847
: an official journal of the American Association for Cancer Research -> delete
L871
: genetics, molecular biology, evolution, endocrinology, embryology, and pathology of sex determination and differentiation -> delete
L892
= journal de l'Association medicale canadienne -> delete
Author Response
Dear authors,
This MS is an interesting and may be important report dealing with negative aspects of 4-O-Methylhonokiol, a major component of magnolia bark extracts, to embryogenesis of a model vertebrate, medaka. Since magnolia bark extract is used as a herbal medicine or diet supplement, and since positive effects of 4-O-Methylhonokiol are recently reported, publication of this MS telling possible unwanted side effects of this herbal compound poses a question on medicinal or supplemental use of this herbal compound to be controlled. It is thus worth publishing of this MS in your journal, if adequately revised. See below for detail.
L9
No one is affiliated with the National Research Center of Egypt.
Answer: It is corrected
L27
Consider adding wards related to Chinese herbal medicine to attract wider readership, though I am not sure if there is a limitation of number of keywords.
Answer: It is corrected
L30
Evidence strongly indicates that (verbose) -> delete
Answer: It is corrected
L32
4-O-Methylhonokiol (MH) -> MH (4-O-Methylhonokiol)
Do not begin a sentence with Arabic numbers. Make a trick to avoid it.
Answer: It is corrected
L34
Magnolias -> magnolias
Answer: It is corrected
L35,48
Magnolia -> {in Italics, or in lower case}
Make sure whether this word is a Latin name or a common name in English. See also L60.
Answer: It is corrected
L42
This 42 suggests that -> delete
Words '.. suggest .. may' make this sentence dubious.
Answer: It is corrected
L50
refers to -> delete
Answer: It is corrected
L51
in common and -> in common, and
Insert a comma.
Answer: It is corrected
L67
coagulations parameters -> {coagulation parameters, parameters of coagulations}
Answer: It is corrected
L68 results
This section is a mess. Method-describing statements frequently interrupt this section. Tell your results straightforward, deleting long interrupting sentences for readers to see your results quickly. I suggest you to consult with the editor for re-organization of your MS into a classical paper structure (introduction - M&M - results - discussion). This is an easy way to clear your result section. If not allowed, write a draft once of the classical structure and then reorder sections simply.
Answer: It is corrected
L69-73
We assessed .. internal concentrations. -> move to the introduction section (L67)
Answer: It is corrected
L75-76
To date, .. development in medaka. -> move to the introduction section
Answer: It is corrected
L76-78
In this study, .. from hour 5 in development (stage 9). -> move to the M&M section or delete
Answer: It is corrected
L79 Fig.1
Replace the bit-mapped figure graphics with a vector data (eps format is recommended). Or otherwise, replace with a graphics with higher resolution (>600 dpi).
This figure is describing your experimental design and suitable for the method section.
Answer: It is corrected and embedded in the method
L84-93
To determine .. stages. -> move to the M&M section or delete
L93
Here, we report that (wordy) -> delete
Answer: It is corrected
L98
I see mortality raise also in 5 uM. You should present the level of significance (p-value) and comparison framework (pairwise or among groups?).
Answer: It is corrected
L111-115
To compare the sensitivity .. at 28 оC for 96 h. -> move to the M&M section or delete
Answer: It is corrected
L115
Exposure of 5 uM MH -> Exposure of 5 uM MH to adults Deletion of prior sentences needs some revision to fit in.
Answer: It is corrected
L121 Fig.2
Replace the bit-mapped figure graphics with a vector data (eps format is recommended). Or otherwise, replace with a graphics with higher resolution (>600 dpi).
L132
not known -> unknown
Answer: It is corrected
L133
percent -> %
Answer: It is corrected
L134-139
To assess .. concentrations. -> move to the M&M section or delete
Some revision of the next sentence may accordingly be necessary to fit with.
Answer: It is corrected
L156-157
First, .. cardiovascular damage. -> move to the M&M section or delete
Answer: It is corrected
L157
This study demonstrated -> We demonstrate here
Answer: It is corrected
L162
Justify this line fully both sides.
Answer: It is corrected
L181
a lower heart rates -> lower heart rates
Answer: It is corrected
L203-209
Investigations .. blood clot. move to the M&M or discussion section or delete
Answer: It is corrected
L209
these three protease -> factor XI (FXI, a protease of intrinsic pathway), factor VII (FVII, a protease of 205 extrinsic pathway), and factor X (FX, a protease of common pathway) coagulation factors
Answer: It is corrected
L218-258
Some parts should be moved to M&M section, while some others to discussion section.
Answer: It is corrected
L223
the fact that endothelium is activated -> possible endothelium activation
You hypothesized it. See L218. It is not a fact.
Answer: It is corrected
L236
Further experiments .. early warning signs. -> delete
Do not excuse here.
Answer: It is corrected
L238
Replace the bit-mapped figure graphics with a vector data (eps format is recommended). Or otherwise, replace with a graphics with higher resolution (>600 dpi).
Box-frame overlaps onto panel A.
Answer: It is corrected
L260-262
Next,.. SOD (Fig. 7). -> move to the M&M section or delete
Answer: It is corrected
L261
FOXO -> forkhead box-containing protein O (FOXO)
Answer: It is corrected
L275
unaffected?
The difference in expression levels is not significant, but ...
Answer: It is corrected
L277,425,457
Evidence indicates that (verbose) -> delete
Answer: It is corrected
L278-280
To learn .. (IL-1beta) (Fig. 7). -> delete or move to M&M section
Answer: It is corrected
L280
these two cytokines -> two cytokines, tissue necrosis factor-279 alpha (TNF-alpha) and interleukin beta (IL-1beta) (in Greek)
Answer: It is corrected
L285-286
However, further investigations .. via MH. -> delete
Do not excuse here. Do it in discussion section.
Answer: It is corrected
L294,303
What does '10 dpf' mean?
L350 says hatching typically takes seven days. Does hatching of MH-treated embryos retard three days?
Answer: It is corrected
L304
significantly obvious -> obvious
Answer: It is corrected
L312-314
Although several transcription factors .. with cardiac dysfunction. -> move to M&M or introduction section, or delete
Answer: It is corrected
L314-316
We analyzed .. and found that -> delete
Answer: It is corrected
L316
their transcript levels -> transcript levels of some key genes in the Wnt/beta-catenin (in Greek) signaling pathway
Answer: It is corrected
L317-319
Since both .. embryos as well. -> move to M&M or introduction section, or delete
Answer: It is corrected
L317
TGF-beta -> transforming growth factor-beta (in Greek)
Answer: It is corrected
L332-333
Next, .. pathway. -> delete
Answer: It is corrected
L336
beta -> in Greek
Answer: It is corrected
L341
Further studies determined that -> delete
Answer: It is corrected
L348-352
Since the fry stages of fish .. at a velocity of >_2 mm/sec. -> move to M&M section or delete
Answer: It is corrected
L353
circadian activity -> response to a light-dark cycle
The word 'circadian' means near-daily.
Answer: It is corrected
L355-360
The lack of clear locomotor activity .. is out of scope of this report.
You may simply say that MH-treated larvae were inactive in comparison with control group, instead of long unclear sentences.
Answer: It is corrected
L362
MH -> MH-treated
Answer: It is corrected
L369
solid close -> solid
Answer: It is corrected
L374
usage in -> usage of
Answer: It is corrected
L376
an effort -> aimed
Answer: It is corrected
L380
usage as supplements in diets and cosmetics
You should mention here or in the conclusion section about routine dose to diet supplements or to herbal medicine in comparison with your experimental dosage.
Answer: It is corrected
L445
In addition, Dvl mRNA was significantly elevated in MH-treated embryos. (redundant repetition of the results) -> delete
This sentence is well summarized in the next sentence as 'overexpression of Dv1'.
Answer: It is corrected
L468
According to the American College of Cardiology, -> give a reference instead
Answer: It is corrected
L503
MH -> of MH
Answer: It is corrected
L506
gm?
Answer: It is corrected
L508
dichloromethane -> dichloromethane (DCM)?
Answer: It is corrected
L688 references
Check carefully the list again from the beginning. Reference list is frequently a mess of errors. Do you follow the journal style? Is numbering and citation in the text consistent? Are author names and their spelling correct? Are journal names correct? Are publication years correct? Is title case for paper titles allowed? Is lower case for journal or book titles allowed?
Answer: It is corrected
L693
: RTP -> delete
Answer: It is corrected
L730
: JAD -> delete
Answer: It is corrected
L733
: an international journal published for the British Industrial Biological Research Association -> delete
Answer: It is corrected
L744
: an official publication of the American Association of Anatomists -> delete
Answer: It is corrected. Please see Ref. 57.
L751
Aquat Toxicol -> Aquatic Toxicology?
Answer: It is corrected. Please see Ref 59.
L810
Circulation. Heart failure -> Circulation: Heart Failure
Answer: It is corrected.
L834
: official journal of the Histochemistry Society -> delete
Answer: It is corrected.
L847
: an official journal of the American Association for Cancer Research -> delete
Answer: It is corrected.
L871
: genetics, molecular biology, evolution, endocrinology, embryology, and pathology of sex determination and differentiation -> delete
Answer: It is corrected.
L892
= journal de l'Association medicale canadienne -> delete
Answer: It is corrected.
Reviewer 2 Report
Medaka and zebrafish are the animals used first for drug toxicity testing due to its high sensitivity to chemical substances. The content of this manuscript is important for the future use of the extract of Magnolia tissue containing 4-O- Methylhonokiol as a supplement and traditional chinese medicine. They analyzed expression of the genes related with heart development by real time qPCR and found that the over expression of Wnt signal pass way was induced by 4-O-Methylhonokiol. But, for the function of Wnt, it is also important not only the expression level but also which site of tissue is expressed. And, it is necessary to add histological expression analysis data by in situ hybridization in the manuscript. Therefore, I recommend that this paper be accepted into MDPI after minor revision. I write my minor comments below.
1. I recommend that authors will describe water temperature in the explanation Figure 1 because medaka developmental speed is sensitive to culture temperature.
2. I feel that the identification of the treatment concentration of MH is unclear because the line graphs overlap in B and D in Figure 2. It is recommended to show the line graph of each treatment concentration by color difference.
3. The treatments of the medaka embryo with MG were done at 26 ℃, but why were the adult treatments done at 28 ℃? Please explain.
4. I did not conform what kind of spinal deformation is occurring, even if I look at the picture of supplement phots. Please show the photography of tissue section demonstrating what kind of spinal deformation was occur in the larvae, which was associated with 5 μM and 10 μM MH in figure 2. And, please add a sentence that explains what form of spinal deformation was.
5. Authors explained that cardiovascular structure, blood flow, and heartbeat was analyzed under microscopy using Cooled Monochrome Camera. Please describe how to calculate the percentage of blood flow reduction and heart beat reduction in methods.
6. B, C pictures in Figure 5 are not clear due to the problem of the depth of field of the microscope. Can you put a false color to these pictures?
7. In Figure 7, authors explained that SOD mRNA was unaffected. But, I think that the SOD2 expression level increased in embryo treated with 10μM HM. Did it mean that mRNA of SOD2 was not degraded and stable?
8. In order to analyze the function of Wnt in the embryos treated with 1.4-O-Methylhonokiol, it is important not only the expression level but also where it is expressed. And, please compare the Wnt expression region between the embryos treated with 1.4-O-Methylhonokiol and control embryo by whole mount in situ hybridization.
9. Authors analyzed the expression levels of TGF-βand Fzd under Wnt/β-catenin signaling pathway in embryos treated with 1.4-O-Methylhonokiol by RT-PCR. But, it is very important for TGF-β and Fzd which tissue region they were expressed. Therefore, please write which tissue region they were expressed in results.
10. Wnt antagonist DKK-1 binds to the same receptor domain of LRP/LRP6 and inhibits Wnt signal. Wnt also binds Fzd. It is important which tissue cells express LRP/LRP6 and Fzd to clarify the relationship between Wnt and DKK-1 whose expression is enhanced in medaka embryos by 1.4-O-Methylhonokiol. And, authors shall analyze which tissue cells express of LRP/LRP6 and Fzd by whole mount in situ hybridization. And, they shall compare these expression patterns in control embryos and 4-O-Methylhonokiol treated embryos.
11. Susceptibility to drugs varies by medaka strain. Therefore, please describe which medaka strain (DRr, ,HdRr, HNI, cab, etc.) you used for experiment in material and methods.
12. Please write the yield rate when isolating MH from seeds of Magnolia grandiflora in methods.
Author Response
Medaka and zebrafish are the animals used first for drug toxicity testing due to its high sensitivity to chemical substances. The content of this manuscript is important for the future use of the extract of Magnolia tissue containing 4-O- Methylhonokiol as a supplement and traditional chinese medicine. They analyzed expression of the genes related with heart development by real time qPCR and found that the over expression of Wnt signal pass way was induced by 4-O-Methylhonokiol. But, for the function of Wnt, it is also important not only the expression level but also which site of tissue is expressed. And, it is necessary to add histological expression analysis data by in situ hybridization in the manuscript. Therefore, I recommend that this paper be accepted into MDPI after minor revision. I write my minor comments below.
1. I recommend that authors will describe water temperature in the explanation Figure 1 because medaka developmental speed is sensitive to culture temperature.
Answer: It is corrected.
2. I feel that the identification of the treatment concentration of MH is unclear because the line graphs overlap in B and D in Figure 2. It is recommended to show the line graph of each treatment concentration by color difference.
Answer: It is corrected. A Figure with better resolution is incorporated.
3. The treatments of the medaka embryo with MG were done at 26 ℃, but why were the adult treatments done at 28 ℃? Please explain.
Answer: All experiments were performed between 26 °C to 28 °C. On the basis of literature, 25-28ºC is an acceptable range (Kinoshita Murata K, Naruse K, Tanaka M. Medaka: biology, management, and experimental protocols. 2009, Wiley-Blackwell).
4. I did not conform what kind of spinal deformation is occurring, even if I look at the picture of supplement phots. Please show the photography of tissue section demonstrating what kind of spinal deformation was occur in the larvae, which was associated with 5 μM and 10 μM MH in figure 2. And, please add a sentence that explains what form of spinal deformation was.
Answer: We agree with the reviewer that spinal deformity should be characterized. We feel that it is out of scope of the manuscript. Here, we wanted to share reproducible observations. However, if the reviewer feels we should delete the statement, we are willing to do so.
5. Authors explained that cardiovascular structure, blood flow, and heartbeat was analyzed under microscopy using Cooled Monochrome Camera. Please describe how to calculate the percentage of blood flow reduction and heart beat reduction in methods.
Answer: Thank you for the comment. The number of embryos with blood flow reduction and blood vessel occlusion have been calculated and expressed as percentage. Heart beats were counted for 1 minute on 3 and 6 dpf for each embryo under the phase contrast microscope (AO Scientific Instruments, Buffalo, NY). It is stated in the manuscript.
6. B, C pictures in Figure 5 are not clear due to the problem of the depth of field of the microscope. Can you put a false color to these pictures?
Answer: A figure with better resolution is embedded in the manuscript.
7. In Figure 7, authors explained that SOD mRNA was unaffected. But, I think that the SOD2 expression level increased in embryo treated with 10μM HM. Did it mean that mRNA of SOD2 was not degraded and stable?
Answer: The statement is rephrased. Please see Fig. 6 (Old Fig. 7) in the revised manuscript.
8. In order to analyze the function of Wnt in the embryos treated with 1.4-O-Methylhonokiol, it is important not only the expression level but also where it is expressed. And, please compare the Wnt expression region between the embryos treated with 1.4-O-Methylhonokiol and control embryo by whole mount in situ hybridization.
Answer: Thank you for the comment. Like the reviewer, we think it is often useful to correlate any given gene expression with that of a known regional or tissue marker gene. Since there was no information about the effect of 4-O-Methylhonokiol on gene expression, we initially obtained the expression pattern of Wnt and related genes in this manuscript. Although it is out of scope of this manuscript, we agree with the reviewer that this idea will be worth pursuing further in the future studies.
9. Authors analyzed the expression levels of TGF-βand Fzd under Wnt/β-catenin signaling pathway in embryos treated with 1.4-O-Methylhonokiol by RT-PCR. But, it is very important for TGF-β and Fzd which tissue region they were expressed. Therefore, please write which tissue region they were expressed in results.
Answer: It is corrected.
10. Wnt antagonist DKK-1 binds to the same receptor domain of LRP/LRP6 and inhibits Wnt signal. Wnt also binds Fzd. It is important which tissue cells express LRP/LRP6 and Fzd to clarify the relationship between Wnt and DKK-1 whose expression is enhanced in medaka embryos by 1.4-O-Methylhonokiol. And, authors shall analyze which tissue cells express of LRP/LRP6 and Fzd by whole mount in situ hybridization. And, they shall compare these expression patterns in control embryos and 4-O-Methylhonokiol treated embryos.
Answer: It is corrected.
11. Susceptibility to drugs varies by medaka strain. Therefore, please describe which medaka strain (DRr, ,HdRr, HNI, cab, etc.) you used for experiment in material and methods.
Answer: We used orange-red variety medaka strain. It is corrected.
12. Please write the yield rate when isolating MH from seeds of Magnolia grandiflora in methods.
Answer:
Answer: MH represents ~ 10% of the crude seed extract. It is corrected
Reviewer 3 Report
In this manuscript, Santu Singha et al showed 4-O-Methylhonokiol (MH) effects on the development of the heart and circulation in addition to known functions in neuronal and immune cells using Japanese medaka as a model. This study indeed provide useful information for the natural product and heart development/circulation research and can be published after some modification.
My main concern is
1. The effect on the heart and circulation is ok, but effect on vasculature is overestatment. The authors need to provide more structure, function and/or molecular markers of vascular defects
Here are are some suggestions and comments for this manuscript.
1. In the figure 2 A, there is a typo (mM) in the X-axis, should be (µm)?
2. In the abstract, the authors mentioned “Overexpression of major proinflammatory mediators and biomarkers of heart and vascular endothelial cell impairment were detected.” This conclusion is overstatement. Higher expression of oxidative stress and/or inflammatory markers could not represent the impairment of vasculature. The authors need to provide more experiment to prove that or re-write their conclusions.
3. the resolution of the figures is not good enough. (Figure 2 and figure 6)
4. the image quality is not good enough in Figure 5 B and C. (video is ok).
5. the vessel occlusion (figure 4) is not clear for me and readers. How to measure it? It seems to be blood accumulation/blockage in my opinion. In addition, does vessel occlution cause by vascular defect? Or problem in heart pumping.
6. To examine the vascular defects during development, I suggest test vascular markers, such as vegfr2, vegfr3, tie2, ephrinb2, EphB4, instead of the markers of thrombosis and vascular tone.
Author Response
In this manuscript, Santu Singha et al showed 4-O-Methylhonokiol (MH) effects on the development of the heart and circulation in addition to known functions in neuronal and immune cells using Japanese medaka as a model. This study indeed provide useful information for the natural product and heart development/circulation research and can be published after some modification.
My main concern is
1. The effect on the heart and circulation is ok, but effect on vasculature is overestatment. The authors need to provide more structure, function and/or molecular markers of vascular defects
Here are are some suggestions and comments for this manuscript.
1. In the figure 2 A, there is a typo (mM) in the X-axis, should be (µm)?
Answer: It is corrected.
2. In the abstract, the authors mentioned “Overexpression of major proinflammatory mediators and biomarkers of heart and vascular endothelial cell impairment were detected.” This conclusion is overstatement. Higher expression of oxidative stress and/or inflammatory markers could not represent the impairment of vasculature. The authors need to provide more experiment to prove that or re-write their conclusions.
Answer: The sentence is revised.
3. the resolution of the figures is not good enough. (Figure 2 and figure 6)
Answer: A new figure is embedded in the manuscript.
4. the image quality is not good enough in Figure 5 B and C. (video is ok).
Answer: A new figure is embedded in the manuscript.
5. the vessel occlusion (figure 4) is not clear for me and readers. How to measure it? It seems to be blood accumulation/blockage in my opinion. In addition, does vessel occlution cause by vascular defect? Or problem in heart pumping.
Answer: Thank you for the comment. As the reviewer is aware, the vessel occlusion or blockage is due to thrombus, accumulation of oxidized fatty acid, endothelial dysfunction/vascular defect, and many more factors. The reviewer is correct. It is blood blockage/clot.
6. To examine the vascular defects during development, I suggest test vascular markers, such as vegfr2, vegfr3, tie2, ephrinb2, EphB4, instead of the markers of thrombosis and vascular tone.
Answer: Thank you for the comment. The reviewer is correct. As the reviewer mentioned, we were interested in thrombosis and vascular tone in order to substantiate the study of the coagulation factors.
Round 2
Reviewer 1 Report
Letter to Authors
Review Report
molecules-417817R1
4-O-Methylhonokiol influences normal cardiovascular development in medaka embryo
Santu K. Singha, Ilias Muhammad, Mohamed Ali Ibrahim, Mei Wang, Nicole M. Ashpole, Zia Shariat-Madar
190123
Dear authors,
Issues that I pointed out have been cleared mostly. A few points remained to be improved. I suggested the editor a minor revision before acceptance. This time errors carelessly made are apparent. Note that there are very few people reading through your MS before publication. Some errors might be overlooked. I actually did it. There will be, on the other hand, many people who read your paper after it is published. Errors, if any, whatever trivial they were, can easily be found by some of the readers. Publishing a paper is a milestone of young scientists' research career especially among your author group. Be extremely careful to check your MS from the beginning before publication to find and fix any errors. See below for detail. I anticipate you can complete a good manuscript for publication.
L34
compounds- -> compounds
L44
evidence also indicates that -> delete
I am sorry I overlooked last time.
L76
While 10 uM MH was not toxic to -> in Roman
L112
Importantly, (wordy) -> delete
L114
providing a strong comparison group -> contrasting clearly with experimental groups
L124
However, a However, -> However,
L126
Since this molecule could not be theoretically formed from MH -> Since {biochemical, metabolic} basis of formation of this molecule from MH is unknown
Formation of any molecules has substantial, not theoretical, bases. Theoretical formation does not make sense.
L127
Regardless -> Nevertheless
L225
What is ROS?
L554
Magnolia -> {in Italics, or in lower case}
Make sure whether this word is a Latin name or a common name in English.
L635 references
Check the list again carefully from the beginning. According to re-organization of the manuscript, renumbering of the list might occur. Are citation in the text and the list consistent?
Author Response
Reviewer !
L34
compounds- -> compounds
Response: Thank you for your critical reading skill. It is corrected
L44
evidence also indicates that -> delete
I am sorry I overlooked last time.
Response: It is corrected
L76
While 10 uM MH was not toxic to -> in Roman
Response: It is corrected
L112
Importantly, (wordy) -> delete
Response: It is corrected
L114
providing a strong comparison group -> contrasting clearly with experimental groups
Response: It is corrected
L124
However, a However, -> However,
Response: It is corrected
L126
Since this molecule could not be theoretically formed from MH -> Since {biochemical, metabolic} basis of formation of this molecule from MH is unknown
Formation of any molecules has substantial, not theoretical, bases. Theoretical formation does not make sense.
Response: It is corrected
L127
Regardless -> Nevertheless
Response: It is corrected
L225
What is ROS?
Response: It is corrected
L554
Magnolia -> {in Italics, or in lower case}
Make sure whether this word is a Latin name or a common name in English.
Response: It is corrected. It is a named after a botanist.
L635 references
Check the list again carefully from the beginning. According to re-organization of the manuscript, renumbering of the list might occur. Are citation in the text and the list consistent?
Response: Citation in text.and the list are consistent